# Lattice-Matched AlInN/GaN/AlGaN/GaN Heterostructured-Double-Channel Metal-Oxide-Semiconductor High-Electron Mobility Transistors with Multiple-Mesa-Fin-Channel Array

**DOI:** 10.3390/ma14195474

**Published:** 2021-09-22

**Authors:** Hsin-Ying Lee, Day-Shan Liu, Jen-Inn Chyi, Edward Yi Chang, Ching-Ting Lee

**Affiliations:** 1Department of Photonics, National Cheng Kung University, Tainan 70101, Taiwan; hylee@ee.ncku.edu.tw; 2Institute of Electro-Optical and Material Science, National Formosa University, Yunlin 63201, Taiwan; dsliu@nfu.edu.tw; 3Department of Electrical Engineering, National Central University, Zhongli 32001, Taiwan; chyi@ee.ncu.edu.tw; 4Department of Materials Science and Engineering, National Yang Ming Chiao Tung University, Hsinchu 30010, Taiwan; edc@mail.nctu.edu.tw; 5Department of Electrical Engineering, Yuan Ze University, Taoyuan 32003, Taiwan

**Keywords:** double-channel metal oxide semiconductor high-electron mobility transistors, Ga_2_O_3_ gate oxide layer, flicker noise, multiple-mesa-fin-channel array, vapor cooling condensation system

## Abstract

Multiple-mesa-fin-channel array patterned by a laser interference photolithography system and gallium oxide (Ga_2_O_3_) gate oxide layer deposited by a vapor cooling condensation system were employed in double-channel Al_0.83_In_0.17_N/GaN/Al_0.18_Ga_0.82_N/GaN heterostructured-metal-oxide-semiconductors (MOSHEMTs). The double-channel was constructed by the polarized Al_0.18_Ga_0.82_N/GaN channel 1 and band discontinued lattice-matched Al_0.83_In_0.17_N/GaN channel 2. Because of the superior gate control capability, the generally induced double-hump transconductance characteristics of double-channel MOSHEMTs were not obtained in the devices. The superior gate control capability was contributed by the side-wall electrical field modulation in the fin-channel. Owing to the high-insulating Ga_2_O_3_ gate oxide layer and the high-quality interface between the Ga_2_O_3_ and GaN layers, low noise power density of 8.7 × 10^−14^ Hz^−1^ and low Hooge’s coefficient of 6.25 × 10^−6^ of flicker noise were obtained. Furthermore, the devices had a unit gain cutoff frequency of 6.5 GHz and a maximal oscillation frequency of 12.6 GHz.

## 1. Introduction

In the past few decades, despite the fact that impressive gallium nitride (GaN)-based depletion- and enhancement-mode single-channel metal-oxide-semiconductor high-electron mobility transistors (MOSHEMTs) are successfully manufactured and widely utilized in various practical systems [1,2,3,4], compelling devices with enhanced performance are still in urgent demand. To enhance performances of GaN-based MOSHEMTs, it is required to increase the electron mobility and sheet electron density of two-dimensional electron gas (2-DEG) channel induced by the polarized AlGaN/GaN heterostructured interface. In general, high Al content in AlGaN barrier layer was explored to increase sheet electron density. However, in addition to the degradation of the associated electron mobility, the epitaxial growth technique of AlGaN layer with high Al content was an extremely difficult challenge [5,6]. Consequently, vertically laminated multiple 2-DEG channels were recently employed [7,8]. However, multiple-hump transconductance (g_m_) characteristics exhibited in the transconductance–gate–source voltage (g_m_–V_GS_) curves due to the effective gate modulation of the multiple channels [7,8,9,10]. Recently, using the structure of multiple-mesa-fin-channel array, the associated enhanced performance of GaN-based MOSHEMTs was demonstrated owing to their superior gate control and heat dissipation [11,12,13]. In this work, to extend the linear transconductance in a wider gate-source voltage range, the multiple-mesa-fin-channel array was used in lattice-matched AlInN/GaN/AlGaN/GaN heterostructured-double-channel MOSHEMTs. Furthermore, because of the inherently advantageous properties of gallium oxide (Ga_2_O_3_) [14,15], and the low density of interface states between Ga_2_O_3_ film and GaN-based semiconductors [16], in this work, a vapor cooling condensation system was employed to deposit it at approximately 80 K as gate oxide layer of MOSHEMTs. In this work, the combination structures of a 30-nm-thick Ga_2_O_3_ gate oxide layer, lattice-matched double channel, and multiple-mesa-fin channel array were simultaneously used for fabricating AlInN/GaN/AlGaN/GaN MOSHEMTs. The associated performances were also measured and analyzed.

## 2. Materials and Growth Methods

The double channel epitaxial layers for fabricating AlInN/GaN/AlGaN/GaN MOSHEMTs were grown on a silicon (Si) substrate by metal-organic chemical vapor deposition (MOCVD, AIXTRON Group, Herzogenrath, Germany). Trimethylgallium (TMG), triethylgallium (TEG), trimethylaluminum (TMA), trimethylindium (TMI), and ammonia (NH_3_) were the precursors. Hydrogen and nitrogen were used as the carrier gases. The designed epitaxial layers, illustrated in Figure 1a, consisted of an AlN nucleation layer (250 nm), a step-graded AlGaN buffer layer (1.1 μm), an undoped GaN buffer layer (1.9 μm), a GaN channel 1 layer (100 nm), an AlN spacer layer (1 nm), an Al_0.18_Ga_0.82_N barrier 1 layer (25 nm), a GaN channel 2 layer (10 nm), an AlN spacer layer (1 nm), an Al_0.83_In_0.17_N barrier 2 layer (8 nm), and a GaN cap layer (2 nm). The AlN and AlGaN layers were grown at 1040–1100 °C in hydrogen ambient using TMG, while the Al_0.83_In_0.17_N layer was grown at 765 °C in nitrogen ambient using TEG in order to incorporate enough indium to the layer so as to achieve the same lattice constant as GaN’s. From the high-resolution transmission electron microscopy (HRTEM) image (JEOL Ltd., Tokyo, Japan) depicted in Figure 1b, the growth thicknesses of the epitaxial layers were similar with the designed thicknesses. Furthermore, as observed in Figure 1b, the channel region shows good matching with low dislocation density. Using a 1-dimensional (1D) Schrödinger-Poisson solver, the simulated band diagram and electron concentration distribution of the epitaxial layers are depicted in Figure 2. The simulation showed that double 2-DEG channels were constructed by the induced polarization in Al_0.18_Ga_0.82_N/GaN interface (channel 1) and the band discontinuity in lattice-matched Al_0.83_In_0.17_N/GaN interface (channel 2). Electron mobility of 1770 cm^2^/V-s and sheet electron density of 1.11 × 10^13^ cm^−2^ in the double channels were obtained using Hall measurement (Ecopia Corp., Anyang, South Korea)at room temperature.

## 3. Device Fabrication

Figure 3 illustrates the 3-dimensional schematic configuration of the studied devices. Prior to patterning strip channel array as multiple-mesa-fin-channels using a He-Cd laser interference photolithography system, the fabrication processes of the devices started with a spread of photoresist AZ6112 on the sample. By adjusting the incident angle of the two-intersected He–Cd laser beams, 500-nm-wide strip channel arrays were patterned. Using a developer to remove the He–Cd laser illuminated photoresist, the patterned photoresist strip channel array was obtained. After depositing laminated metals of Ni/Au (20/100 nm) with an electron-beam evaporator, the Ni/Au metal mask was formed by lifting off the remaining Ni–Au metals above the photoresist strip channels. To fabricate the multiple-mesa channel, the unmasked region of the sample was etched down to the GaN buffer layer using a photoelectrochemical (PEC) etching method. The etching process and technology of the PEC etching method were demonstrated and reported previously [17]. Under a patterned Ni metal mask (500 nm), the mesa isolation region with an area of 310 μm × 320 μm was formed by etching down to the Si substrate using BCl_3_ etchant in a reactive-ion etching system. To completely remove the undesired native oxide residing on the surface of GaN cap layer, the sample was then surface-treated using an (NH_4_)_2_S_x_ chemical solution at 60 °C for 30 min. The (NH_4_)_2_S_x_ surface-treatment method was previously demonstrated and reported [18]. The source electrode and drain electrode of Ti/Al/Pt/Au (25/100/50/300 nm) laminated multiple metals were deposited using an electron-beam evaporator and then thermally annealed in a rapid-thermal-annealing system under a nitrogen atmosphere at 850 °C for 1 min. The separation between source electrode and drain electrode was approximately 10 μm. Prior to using a vapor cooling condensation system to deposit a 30 nm thick Ga_2_O_3_ gate oxide layer at approximately 80 K under liquid nitrogen cooling, the surface-treatment technique of a (NH_4_)_2_S_x_ chemical solution was utilized to treat the sample again. The deposition processes and performance of Ga_2_O_3_ films deposited by the vapor cooling condensation system were previously demonstrated and reported [19,20]. Using a standard photolithography method to pattern two-finger gate regions, Ni/Au (20/300 nm) gate laminated metals were deposited using the electron-beam evaporator, and the two-finger Ni/Au gate metals were manufactured using a lift-off process. Gate width and length were 50 and 1 μm, respectively. Furthermore, Ni/Au gate metals were placed in the central regions between source electrode and drain electrode.

## 4. Results and Discussion

Figure 4 depicts the HRTEM image of the fin-channel. From the observation of HRTEM image, the height, width, and spacing of the multiple-mesa-channel were 69.8 nm, 496.6 nm, and 490.6 nm, respectively. Consequently, the total real channel width within a 50-μm-wide gate region was about 25.2 μm. By applying various gate-source voltage (V_GS_) levels, typical drain-source current (I_DS_)–drain-source voltage (V_DS_) characteristics of the studied devices, shown in Figure 5, were obtained using the measurement of an Agilent 4156C semiconductor parameter analyzer. Under the operating voltage of V_DS_= 10 V, its normalized saturation drain-source current was 352.0 and 842.7 mA/mm of the devices at V_GS_ = 0 V and 5 V, respectively. At the operating voltage of V_DS_ = 10 V, the dependence of drain-source current and extrinsic transconductance (g_m_) on gate-source voltage are depicted in Figure 6. Maximal extrinsic transconductance was 148.9 mS/mm. In general, the double-hump transconductance behaviors were caused by the effective gate modulation of the upper and lower channels in the double-channel MOSHEMTs, respectively [8,9,10]. However, double-hump transconductance characteristics did not appear in the g_m_–V_GS_ curve of the studied devices. In the studied devices, the wider smooth g_m_ value distribution caused by the collapse paving of double-hump transconductance behavior was contributed to the superior gate control of fin-channel. The superior gate control was attributed to the side-wall electric field modulation in the fin channel. By defining the on-resistance (R_on_) as the inverse slope of the I_DS_–V_DS_ characteristics at V_GS_ = 5 V and V_DS_ = 0 V, the associated on-resistance of 6.1 Ω-mm was obtained. Furthermore, when threshold voltage (V_th_) was defined as the gate-source voltage corresponding to the drain-source current of 1 μA/mm, V_th_ was −3.2 V. To measure high-frequency performance of the studied devices using an Agilent 8510C network analyzer, Figure 7 illustrates the small-signal high-frequency performance of frequency-dependent short-circuit current gain and maximal available power gain. Figure 7 shows that the unit gain cutoff frequency (f_T_) and maximal oscillation frequency (f_max_) were 6.5 and 12.6 GHz, respectively. In general, to evaluate electron trapping and electron detrapping behaviors induced from defects, traps, and interface states residing in electronic devices, the measurement of low-frequency noise performance was effective [21]. Under the operation of V_DS_ = 1 V, Figure 8 depicts the frequency (f)-dependent normalized noise power density spectra ($S_{I_{\mathrm{DS}}}\left(f\right)/I_{\mathrm{DS}}{}^{2}$) of the studied devices measured by an Agilent 4156C semiconductor analyzer, an HP 35670A dynamic signal analyzer, and a BTA 9812B noise analyzer. The normalized noise power density gradually decreased with an increase in gate-source voltage. Due to the quite good variation between normalized noise power density and 1/f, flicker noise was the dominant noise of the devices. At the operating condition of f = 10 Hz, V_DS_ = 1 V, and V_GS_ = 5 V, the normalized noise power density was approximately 8.7 × 10^−14^ Hz^−1^. Using a mobility fluctuation model [22], Hooge’s coefficient α, a useful figure-of-merit parameter, could be expressed as:(1)$$\alpha =\left(S_{I_{\mathrm{DS}}}\left(f\right)/I_{\mathrm{DS}}{}^{2}\right)\cdot f\cdot \left(L_{G}W_{G}n_{\mathrm{ch}}\left(V_{\mathrm{GS}}-V_{\mathrm{th}}\right)/\left|V_{\mathrm{th}}\right|\right)$$
where L_G_ =1 μm and W_G_ = 25.2 μm are the gate length and real gate width on multiple-mesa-fin-channels, respectively. The n_ch_ = 1.11 × 10^13^ cm^−2^ is the sheet electron density of the double-channel and V_th_ = −3.2 V is the threshold voltage. By substituting those parameters into Equation. 1, the α value of 6.25 × 10^−6^ was calculated for devices operating at f = 10 Hz, V_DS_ = 1 V, and V_GS_ = 5 V.

## 5. Conclusions

In this work, a MOCVD system was utilized to grow Al_0.83_In_0.17_N/GaN/Al_0.18_Ga_0.82_N/GaN heterostructured-epitaxial layers on Si substrates. According to the simulated energy band structure, polarization in Al_0.18_Ga_0.82_N/GaN interface and the band discontinuity in lattice-matched Al_0.83_In_0.17_N/GaN interface induced channels 1 and 2, respectively. The resulting double-channel epitaxial layers and multiple-mesa-fin-channel array were utilized for fabricating MOSHEMTs. Benefittig from the better gate control capability caused by the modulation of side-wall electrical field in fin-channel, the double-hump transconductance behavior, which occurred in double-channel MOSHEMTs, was not obtained. The multiple-mesa-fin-channel array used in double-channel MOSHEMTs could effectively pave the collapse of transconductance due to their better gate control capability. Owing to the high-insulating Ga_2_O_3_ gate oxide layer deposited by the vapor cooling condensation system and the inherent high-quality interface between the Ga_2_O_3_ and GaN layers, the low-flicker-noise performance was achieved. Under the operation of f = 10 Hz, V_GS_ = 5 V, and V_DS_ = 1 V, the low Hooge’s coefficient α was approximately 6.25 × 10^−6^.

## Figures and Tables

**Figure 1 materials-14-05474-f001:**
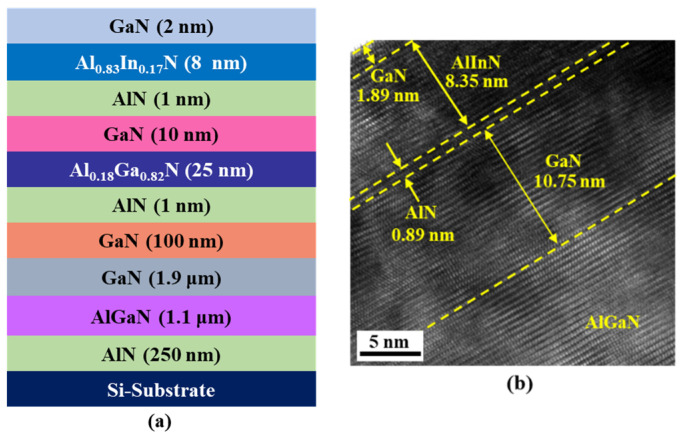
(**a**) Schematic structure and (**b**) high-resolution transmission electron microscopy image of epitaxial layers.

**Figure 2 materials-14-05474-f002:**
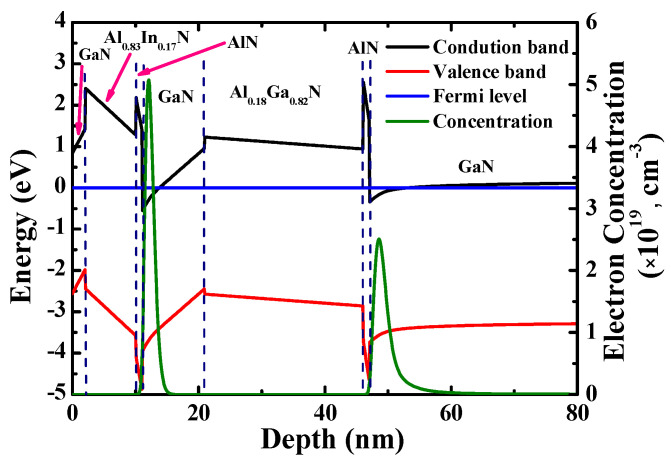
Simulated band diagram and electron concentration distribution in epitaxial layers.

**Figure 3 materials-14-05474-f003:**
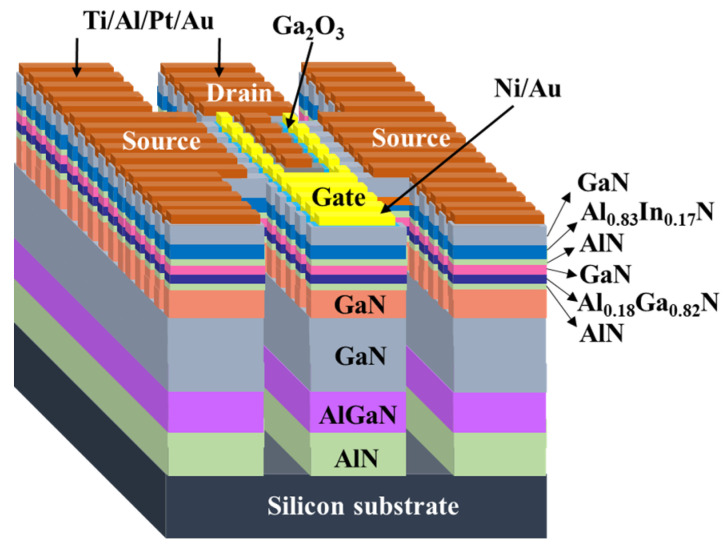
Three-dimensional schematic configuration of MOSHEMTs with multiple-mesa-fin-channel array.

**Figure 4 materials-14-05474-f004:**
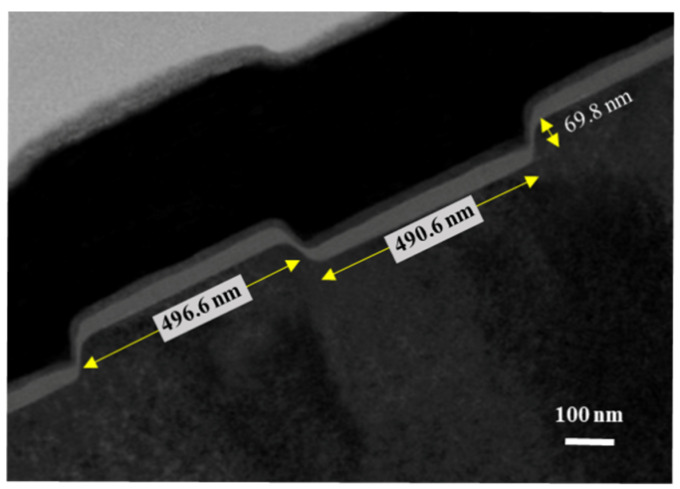
High resolution transmission electron microscope image of cross-sectional fin-channel.

**Figure 5 materials-14-05474-f005:**
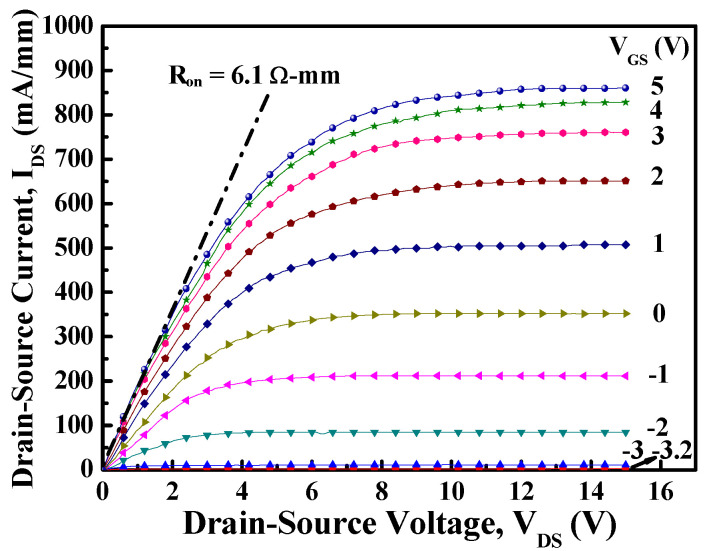
Typical drain−source current−drain−source voltage characteristics of MOSHEMTs with multiple-mesa-fin-channel array.

**Figure 6 materials-14-05474-f006:**
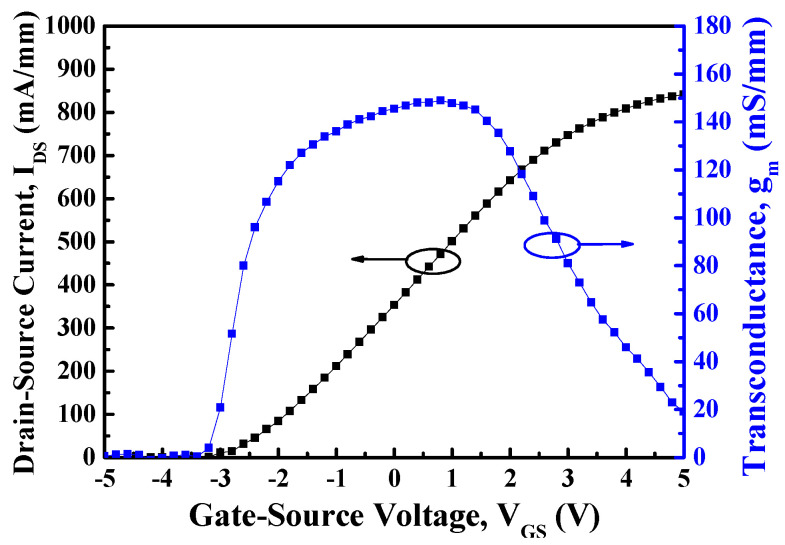
Under the operation of drain−source voltage of 10 V, drain−source current and extrinsic transconductance as a function of gate−source voltage of MOSHEMTs with multiple-mesa-fin-channel array.

**Figure 7 materials-14-05474-f007:**
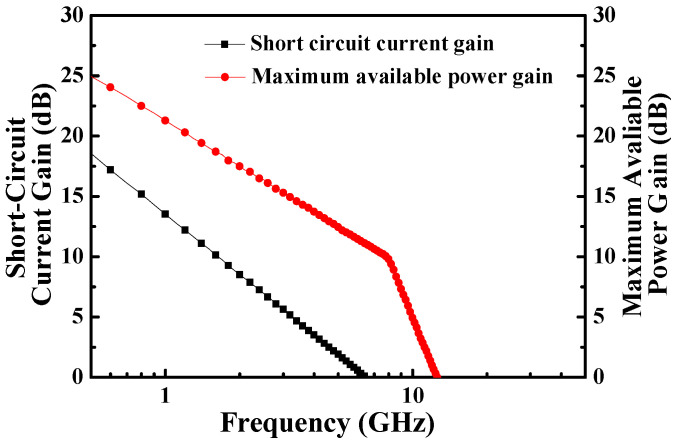
Short-circuit current gain and maximum available power gain as a function of frequency of MOSHEMTs with multiple-mesa-fin-channel array.

**Figure 8 materials-14-05474-f008:**
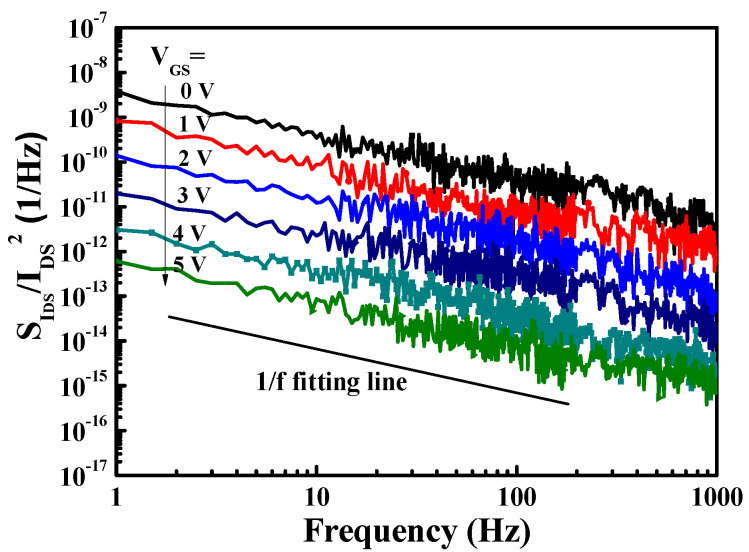
Under drain−source voltage of 1 V, frequency-dependent normalized noise power spectra of MOSHEMTs with multiple-mesa-fin-channel array.

## Data Availability

The data presented in this study are available on request from the corresponding author.

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
