# Peer review of "Lattice-Matched AlInN/GaN/AlGaN/GaN Heterostructured-Double-Channel Metal-Oxide-Semiconductor High-Electron Mobility Transistors with Multiple-Mesa-Fin-Channel Array"

_materials, 2021, doi:10.3390/ma14195474_

Round 1

Reviewer 1 Report

General Remarks:

The manuscript provides exciting data about the Multiple-mesa-fin-channel array and gallium oxide (Ga2O3) gate oxide layer. Overall, the paper is well written; however, it is necessary to improve the presentation.

The authors address a topic that many authors have already studied, so it is necessary to show the innovation of this work. This approach should be made in the "Abstract and Introduction." By the way, the introduction should be rewritten entirely to make the manuscript more attractive.

The materials and methods section should be reworded. I suggest that the authors better define how each layer was grown. This section could be divided into two subsections: i) materials and growth methods and ii) characterization techniques used.

The results and conclusion section needs better explored, showing the innovation of the present work. I also see the need for a better literature review, as references are poor.

Although the work is interesting – a major revision should be carried out before publication.

Reviewer 2 Report

Authors present some interesting results in the field of double-channel MOSHEMT’s structures. Presented scientific investigation is well presented, however some improvements can be done:

  • The work should be linguistically corrected, there many errors, like “To add features of …” (pg. 1 l. 35), “the associated epitaxial growth technique…” (pg. 1 l. 39), “low interface states..” (pg. 2 l. 52) instead of “low density of interface states” and many others. I suggest a thorough editing of the text.
  • In the title, authors claim “lattice-matched …. heterostructured….”, however, there is no comment in the text about lattice matching of subsequent epitaxial layers to each other and to the substrate. Provided TEM image of the channel region shows good matching with low dislocation density, however some comments should be added in the text.
  • Figure 2 shows band diagram of the simulated structure and the electron concentration distribution in heterostructure. Please change the caption of the figure to “Simulated bad diagram and electron concentration distribution in ….” Please also change the left axis title to “Energy”, not “Energy level” and right axis title to “Electron concentration”.
  • It would be advisable to present the SEM images of fabricated structures with mesas.
  • Authors claim, that the width and period of mesa patterning was 496.6 nm and 982.6 nm respectively. Are these average values? In general, such accuracy of measurement is not necessary in this case.
  • I suggest to add IDS, VDS, VGS and gm acronyms in addition to “drain-source current”, “drain-source voltage”, “gate-source voltage” and “transconductance” in figures 4 and 5 .

Reviewer 3 Report

In this paper the authors investigated Multiple-mesa-fin-channel array and gallium oxide (Ga2O3) gate oxide layer were employed in double-channel Al0.83In0.17N/GaN/Al0.18Ga0.82N/GaN heterostructured-metal-oxide-semi-17 conductors (MOSHEMTs). The objective of the work is well determined. The text is clearly written. Materials and methods are well described. The flicker noise was deduced to be the dominate noise of the devices. Under the operation of f = 10 Hz, VGS = 5 V, and VDS = 1 V, the low Hooge’s coefficient alfa (a useful figure-of-merit parameter) was approximately 6.25x10^-6. The References are sufficient and appropriate. The work is interesting and should be accepted by the journal.

Round 2

Reviewer 1 Report

The authors made all the changes requested by the reviewer. Therefore I agree with the publication of the article in its present form.